# A New Paradigm for Federated Structure Non-IID Subgraph Learning

## Abstract

Federated graph learning (FGL), a distributed training framework for graph neural networks (GNNs) has attracted much attention for breaking the centralized machine learning assumptions. Despite its effectiveness, the differences in data collection perspectives and quality lead to the challenges of heterogeneity, especially the domain-specific graph is partitioned into subgraphs in different institutions. However, existing FGL methods implement graph data augmentation or personalization with community split which follows the cluster homogeneity assumptions. Hence we investigate the above issues and suggest that subgraph heterogeneity is essentially the structure variations. From the observations on FGL, we first define the structure non-independent identical distribution (Non-IID) problem, which presents unique challenges among client-wise subgraphs. Meanwhile, we propose a new paradigm for general federated data settings called Adaptive Federated Graph Learning (AdaFGL). The motivation behind it is to implement adaptive propagation mechanisms based on federated global knowledge and non-params label propagation. We conduct extensive experiments with community split and structure Non-IID settings, our approach achieves state-of-the-art performance on five benchmark datasets.

## 1 Introduction

The graph as a relational data structure is widely used to model real-world entity relations such as citation networks Yang et al. (2016a), recommended systems Wu et al. (2022), drug discovery Gaudelet et al. (2021), particle physics Shlomi et al. (2021), etc. However, due to the collection agents and privacy concerns, generally, the global domain-specific graph consists of many subgraphs collected by multiple institutions. In order to analyze the local subgraph, each client maintains a powerful graph mining model such as graph neural networks (GNNs), which have achieved state-of-the-art performance in many graph learning tasks Zhang et al. (2022b); Hu et al. (2021); Zhang & Chen (2018). Despite its effectiveness, the limited data provide sub-optimal performance in most cases. Motivated by the success of federated learning (FL), a natural idea is to combine the GNNs with FL to utilize the distributed subgraphs. Recently, federated graph learning (FGL) He et al. (2021); Wang et al. (2022b) is proposed to achieve collaborative training without directly sharing data, yet an essential concern is the heterogeneity of the distributed subgraphs.

Notably, graph heterogeneity is different from the heterogeneity of labels or features in the fields of computer vision or natural language processing, we suggest that it depends on the graph structure. However, The existing FGL methods simulate the federated subgraph distributions through community split, which follows the cluster homogeneity assumption as shown in Fig.1(a). Specifically, community split leads to the subgraph structure being consistent and the same as the original graph, e.g., connected nodes are more likely to have the same labels. Obviously, it is overly desirable and hard to satisfy in reality, hence we consider a more reasonable setting shown in Fig.1(c). We first refer to the above problem as structure non-independent identical distribution (Non-IID).

The motivation behind it is due to graph structure directly related to node labels and feature distributions. Meanwhile, the challenges of structure heterogeneity are ubiquitous in the real world Zheng et al. (2022b). For instance, in citation networks, we consider research teams focused on computers and intersectional fields (e.g., AI in Science) Shlomi et al. (2021); Gaudelet et al. (2021) as clients. In online transaction networks, fraudsters are more likely to build connections with customers instead

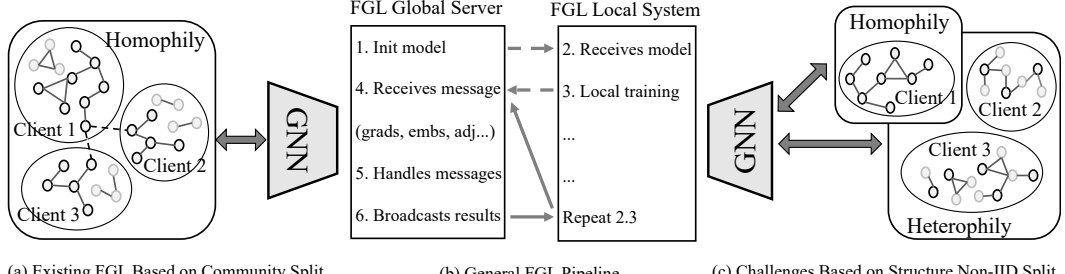

(a) Existing FGL Based on Community Split     (b) General FGL Pipeline     (c) Challenges Based on Structure Non-IID Split

Figure 1: We utilize black circles for base class and gray circles for other class. (a): Limitations of existing FGL methods. (b): The general collaborative training pipeline. (c): The new challenges of graph structure Non-IID for FGL.

of other fraudsters Pandit et al. (2007). We consider different regions as clients to detect financial fraudsters by analyzing online transaction subgraphs. Specifically, graph structure can be divided into two types: homogeneity means that connected nodes are more likely to have the same label and similar feature distributions and heterogeneity is the opposite. In order to explain it intuitively, we visualize the 3 clients partitioning result on Cora in Table. 1 and Table. 2, where Homo represents the homogeneity degree of the local subgraph, and it is computed by a popular metric Pei et al. (2020). Obviously, compared to community split, which follows the cluster homogeneity assumption and uniform distribution principle, structure Non-IID brings challenges to the existing FGL methods.

Table 1: Community split in Cora.

| Community | #Nodes | #Edges | Homo |
|---|---|---|---|
| Client1 | 903 | 1696 | 0.85 |
| Client2 | 903 | 1575 | 0.78 |
| Client3 | 902 | 1592 | 0.87 |

Table 2: Structure Non-IID in Cora.

| Non-IID | #Nodes | #Edges | Homo |
|---|---|---|---|
| Client1 | 1095 | 1473 | 0.43 |
| Client2 | 946 | 1400 | 0.87 |
| Client3 | 667 | 1212 | 0.31 |

Based on this, we investigate the above issues through empirical analysis shown in Fig. 2. According to the results, we observe that in case the original graph satisfies the homogeneity assumption then the label distributions satisfy Non-IID. It is the opposite when the original graph satisfies the heterogeneity. This is due to the fact that the nodes partitioned into the same clients are communities and follow the uniform distribution principle. In addition, the local accuracy indicates that the subgraph structure performs a more important role in FGL compared to the label distributions, which also supports our motivation. In model performance, we observe that the GGCN improves the structure Non-IID problem, and FedSage+ trains NeighGen to implement local subgraph augmentation by sharing node embeddings. However, the above methods fail to achieve competitive results as SGC on the homogeneous subgraphs while considering heterogeneity.

In order to efficiently analyze distributed subgraphs with both homogeneity and heterogeneity. We propose a simple pipeline called Adaptive Federated Graph Learning (AdaFGL) for more general federated data settings, which consists of three main parts. Specifically, it starts by analyzing the subgraph structure through non-params label propagation and selects the appropriate base model: (i) the federated global knowledge extractor (e.g., MLP, powerful GNNs, or any reasonable embedding models), which does not rely on any learning over the subgraph. Then, the base predictor is trained based on the global data, which can be done offline or in parallel with local training, benefiting from the flexibility of our approach. Finally, the local client implements two adaptive propagation mechanisms: (ii) homogeneity propagation module or (iii) heterogeneity propagation module based on the local subgraph. Notably, with non-params label propagation, the above process is adaptive.

To summarize, the contributions of this paper are as follows: (1) To the best of our knowledge, we are the first to analyze the structure Non-IID problem in FGL, which is a more general federated data setting and brings new challenges. (2) We propose AdaFGL, a new paradigm for structure Non-IID subgraph learning, which shows its flexibility in FGL with impressive performance. (3) Extensive experiments demonstrate the effectiveness of AdaFGL. Specifically, our approach achieves state-of-the-art performance in the above two data settings. Compared to the best prediction accuracy in the

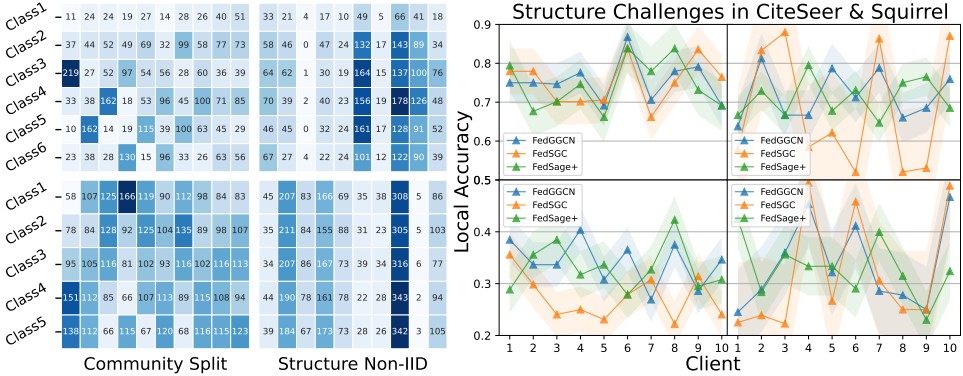

Figure 2: FedSGC: FedAvg+SGC, FedGGCN: FedAvg+GGCN represent the representative methods. The upper and lower parts represent CiteSeer and Squirrel, respectively, left and right parts represent community split and structure Non-IID, respectively. The left x-axis represents the client ID, and the number represents the number of samples in each client corresponding to the classes. The dark color represents the number of samples with high numbers.

baselines, our method achieves performance gains of $4.67\%$ and $2.65\%$ in structure Non-IID and community split data settings, respectively.

## 2 PRELIMINARIES

In this section, we first introduce the semi-supervised node classification task. Then, we review the prior diverse GNNs and very recent FGL methods. Consider a graph $G = (V, E)$ with $|V| = $ n nodes and $|E| = $ m edges, the adjacency matrix (including self loops) is denoted as $\hat{\mathbf{A}} \in \mathbb{R}^{n \times n}$, the feature matrix is denoted as $\mathbf{X} = \{x_1, x_2, \ldots, x_n\}$ in which $x_v \in \mathbb{R}^f$ represents the feature vector of node $v$, and $f$ represents the dimension of the node attributes. Besides, $\mathbf{Y} = \{y_1, y_2, \ldots, y_n\}$ is the label matrix, where $y_v \in \mathbb{R}^{|Y|}$ is a one-hot vector and $|Y|$ represents the number of the node classes. The semi-supervised node classification task is based on the topology of labeled set $V_L$ and unlabeled set $V_U$, and the nodes in $V_U$ are predicted with the model supervised by $V_L$.

**GNNs.** As the most popular GNN method, The forward information propagation process of the $l$-th layer GCN Kipf & Welling (2017) is formulated as

$$\mathbf{X}^{(l)} = \sigma(\tilde{\mathbf{A}}\mathbf{X}^{(l-1)}\mathbf{W}^{(l)}), \ \tilde{\mathbf{A}} = \hat{\mathbf{D}}^{r-1}\hat{\mathbf{A}}\hat{\mathbf{D}}^{-r}, \tag{1}$$

where $\hat{\mathbf{D}}$ represents the degree matrix with $\hat{\mathbf{A}}$, $r \in [0, 1]$ denotes the convolution kernel coefficient, $\mathbf{W}$ represents the trainable weight matrix, and $\sigma(\cdot)$ represents the non-linear activation function. In GCN, we set $r = 1/2$, and then $\hat{\mathbf{D}}^{-1/2}\hat{\mathbf{A}}\hat{\mathbf{D}}^{-1/2}$ is called symmetric normalized adjacency matrix. Despite their effectiveness, they have limitations in real-world graphs, which have complex heterogeneous relationship patterns. Some recent researches Liu et al. (2021); Chien et al. (2021); He et al. (2022); Wang et al. (2022a); Yang et al. (2022) solve it by higher-order neighborhood discovery or message combination strategies to improve the GNN process via

$$\begin{aligned} \mathbf{m}_v^{(l)} &= \mathtt{Aggregate}^{(l)}(\{\mathbf{h}_u^* | u \in \mathcal{N}_*(v)\}), \\ \mathbf{h}_v^{(l)} &= \mathtt{Update}^{(l)}(\mathbf{h}_v^*, \mathbf{m}_v^*), \end{aligned} \tag{2}$$

where $\mathbf{h}_u^*$ denotes the information of multi-hop neighbors $\mathcal{N}_*(v)$, $\mathbf{m}_v^*$ represents the higher-order messages of node $v$ from the previous layers, $\mathtt{Aggregate}(\cdot)$ and $\mathtt{Update}(\cdot)$ denote the message aggregation function and update function, respectively. However, these methods suffer from high computational complexity and fail to achieve competitive performance on the homogeneous graph.

**FGL** has received growing attention for breaking centralized graph machine learning assumptions. FedGraphNN He et al. (2021) and FS-G Wang et al. (2022b) propose general FGL packages, which contain a wide range of graph learning tasks. GCFL Xie et al. (2021) and FED-PUB Baek et al. (2022) investigate the personalized technologies in graph-level and node-level, respectively. Furthermore, some recent researches improve performance with local subgraph augmentation, including FedGNN Wu et al. (2021), FedGL Chen et al. (2021), and FedSage Zhang et al. (2021). Inspired

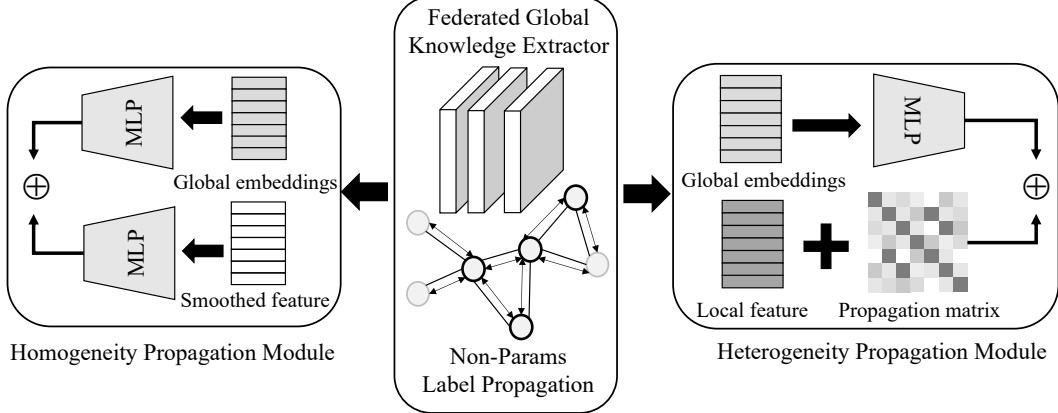

Figure 3: Overview of our model-free pipeline with a toy example. The federated global knowledge extractor represents a wide variety of node embedding models. The middle part of the black circles represents labeled nodes and the gray circles represent unlabeled nodes. Based on this we obtain graph structure properties and implement the adaptive propagation mechanisms.

by FS-G Wang et al. (2022b), we can consider the collaborative training process in FGL as modules. Specifically, we model the information such as gradients and node embeddings uploaded by the clients as messages. Then we consider the server processes and broadcast results as the various message-handling mechanisms. Here we illustrate the GNNs combined with collaborative training. Its generic form with $N$ clients is defined as

$$\texttt{FGL} - \texttt{Clients}\,(\texttt{Local Update}) \to \min \frac{1}{N} \sum_{i=1}^{N} \mathbb{E}_{(\mathbf{A}_i, \mathbf{X}_i, \mathbf{Y}_i) \sim \mathcal{D}_i} [\mathcal{L}_{ce}(f_{\theta_i}(\mathbf{A}_i, \mathbf{X}_i), \mathbf{Y}_i)],$$

$$\mathcal{L}(f_{\theta_i}(\mathbf{A}_i, \mathbf{X}_i), \mathbf{Y}_i) = -\sum_{i \in V_L} \sum_{j} [\mathbf{Y}_{ij} \log(\texttt{Softmax}(\tilde{\mathbf{Y}}_{ij})) + (1 - \mathbf{Y}_{ij}) \log(1 - \texttt{Softmax}(\tilde{\mathbf{Y}}_{ij}))],$$

(3)

where $f_{\theta_i}$ and $\mathcal{L}_{ce}$ are the $i$-th local GNN with parameters $\theta$ and cross-entropy loss function, respectively. It can be replaced by any other appropriate loss function depending on the task. $(\mathbf{A}_i, \mathbf{X}_i, \mathbf{Y}_i) \sim \mathcal{D}_i$ represents the local subgraph $(\mathbf{A}_i, \mathbf{X}_i, \mathbf{Y}_i)$ sampled from the distribution $\mathcal{D}_i$. FedAvg McMahan et al. (2017) is an efficient FL algorithm, which can be defined as

$$\texttt{FGL} - \texttt{Server}\,(\texttt{Aggregate}) \to \forall i, \mathbf{W}_i^{t+1} \leftarrow \mathbf{W}_i^t - \eta g_i, \; \mathbf{W}^{t+1} \leftarrow \sum_{i=1}^{N} \frac{\mathrm{n}_i}{\mathrm{n}} \mathbf{W}_i^{t+1},$$

(4)

where $t$ represents the round number of the FL process, $\mathbf{W}$ represents the model weights, $\eta$ represents the learning rate, $g$ represents the gradient calculated from the Eq. 3, $\mathrm{n}_i$ and $\mathrm{n}$ represent the $i$-th local client data size and the global data size, respectively.

## 3 ADAFGL PIPELINE

The basic idea of AdaFGL is to perform adaptive propagation mechanisms based on federated global knowledge and non-params label propagation. The pipeline with three main parts as shown in Fig. 3, which combine the global knowledge embeddings and local structure properties. The above decoupling process utilizes the computational capacity of the local system while minimizing communication costs and the risk of privacy leakage. AdaFGL can benefit from the evolution of FL and GNN through the base predictor and adaptive propagation. Notably, the base predictor obtained by federated training and personalized propagation are viewed as two decoupled modules that are executed sequentially. Meanwhile, both of them accomplish the training without sharing local private data.

### 3.1 FEDERATED GLOBAL KNOWLEDGE EXTRACTOR

In FGL, limited data yields sub-optimal performance in most cases. Therefore, AdaFGL starts to perform non-params label propagation to adaptive process. Note this process does not rely on any

learning over the subgraph. Specifically, the labeled nodes are initialized as $\mathbf{y}_v^0 = y_v, \forall v \in V_L$, and the unlabeled nodes are denoted as $\mathbf{y}_u^0 = (\frac{1}{|Y|}, \ldots, \frac{1}{|Y|}), \forall u \in V_U$. Then, the non-params label propagation of the $k$-step is expressed as

$$
\begin{aligned}
\mathbf{y}_u^k &= \texttt{graph} - \texttt{aggregator}(\{\mathbf{y}_v^{k-1} | v \in \mathcal{N}_u\}) \\
&= \alpha \mathbf{y}_u^0 + (1-\alpha) \sum_{v \in \mathcal{N}_u} \frac{1}{\sqrt{\tilde{d}_v \tilde{d}_u}} \mathbf{y}_v^{k-1}.
\end{aligned} \tag{5}
$$

We follow the approximate calculation of the personalized PageRank matrix Klicpera et al. (2019), where $\mathcal{N}_v$ represents the one-hop neighbors of $v$, and we default set $\alpha = 0.5$. Then, we design the homogeneity confidence score (HCS) computed by the number of correct predictions, and the default ratio of the boolean mask is 0.5. Finally, we set thresholds $\lambda$ for the adaptive binary selection of the homogeneity propagation module and heterogeneity propagation module in each client. In experiments, we default set $\lambda = 0.6$ To demonstrate that AdaFGL is a simple yet effective framework, we choose simple models (e.g. MLP or SGC) and FedAvg to achieve federated training. Due to the flexibility of AdaFGL, they can be replaced by any other powerful GNNs and federated methods.

From the perspective of FL in Non-IID data, we default choose MLP as the base predictor, which is independent of the graph structure. Then we quote the convergence theorem Li et al. (2020) in $T$ rounds and $E$ epochs, the federated global knowledge extractor error bound $\epsilon_{fed}$ is expressed as

$$
\epsilon_{fed} \leq \frac{2L}{\mu^2(\gamma + T - 1)} \left( \sum_{i=1}^N \frac{\mathrm{n}_i}{\mathrm{n}} \varphi_i^2 + 6L\phi + 8(E-1)^2\omega^2 + \frac{\gamma}{4} ||\mathbf{W}_1 - \mathbf{W}^\star||^2 \right). \tag{6}
$$

It assumes that the mapping function satisfies $L$-smooth and $\mu$-strongly convex, where $\varphi$ and $\phi$ represent the local random gradient and the degree of model heterogeneity, respectively, $\gamma = \max\{8L/\mu, E\}$, $\omega$ denotes the divergence of local model, and $\mathbf{W}^*$ represents the global optima.

We observe that the base predictor error bound is mainly determined by the differences in the node feature distributions, and the model performance will be further hurt if the graph structure is considered. Therefore, we are motivated to propose adaptive propagation mechanisms. Specifically, we implement the binary selection of the homogeneity propagation module or heterogeneity propagation module in each client by comparing the HCS value and the threshold $\lambda$. We will describe the technical details of personalized propagation strategies.

## 3.2 Adaptive Homogeneity Propagation

After that, we use the base predictor to embed local subgraph nodes into the global knowledge space $\mathbf{X}_{global}$ and improve the accuracy with the local homogeneous structure. The motivation behind it is that the feature propagation satisfying homogeneity has a significant positive impact on prediction performance, which has also been confirmed in many recent research works Zhang et al. (2022a); Wang & Leskovec (2020). Hence we expect to utilize local smoothing features to correct the predictions. Then, we first define the homogeneous feature propagation

$$
\begin{aligned}
\mathbf{X}_{smooth}^{(k)} &= \texttt{graph} - \texttt{operator}(\mathbf{A})^{(k)} \mathbf{X}^{(0)}, \forall k = 1, \ldots K, \\
\mathbf{H}_{homo} &= \texttt{message} - \texttt{updater}(\mathbf{X}_{smooth}^{(k)}) = f_\theta(\mathbf{X}_{smooth}^{(k)}),
\end{aligned} \tag{7}
$$

where $\texttt{graph} - \texttt{operator}(\cdot)$ represents the graph operator in feature propagation, we default to use symmetric normalized adjacency as shown in Eq. 1. $\mathbf{X}_{smooth}^{(k)}$ represents the local smoothing features after $K$-steps propagation, $\texttt{message} - \texttt{updater}(\cdot)$ denotes the model training process, and we use $f_\theta$ to represent the linear regression or MLP with parameters of $\theta$.

In order to correct the global embedding and local information, we use the local message update mechanism and online distillation to achieve an effective combination of the local smooth structure prior and the global embeddings, which can be written as

$$
\begin{aligned}
\mathbf{H}_{local} &= \mathbf{W}_{local} \mathbf{X}_{global}, \\
\mathcal{L}_{kd} &= ||\mathbf{H}_{homo} - \mathbf{H}_{local}||_F.
\end{aligned} \tag{8}
$$

Based on this, we can make local smoothing information and global embeddings to achieve mutual supervision and end-to-end training by gradient updating. This exploits the local structure information to reduce the error bound. Notably, the above adaptive process is accomplished in the local client and has no additional communication costs and privacy concerns.

### 3.3 Adaptive Heterogeneity Propagation

In contrast, in order to break the heterogeneous structure limitations, we optimize the message-passing framework by embeddings $\mathbf{X}_{global}$ to detect subgraph heterogeneous patterns. Specifically, we propose an adaptive propagation mechanism by discovering the global dependency of the current node and modeling the positive or negative impact of the messages. Intuitively, we first expect to optimize the propagation probability matrix and align the local structure by global embeddings

$$
\mathbf{A}_{prop}^{(0)} = \mathbf{X}_{global}\mathbf{X}_{global}^{T},
$$
$$
\mathbf{X}_{align} = \texttt{graph} - \texttt{operator}(\hat{\mathbf{A}}_{prop}^{(0)})^{(k)}\mathbf{X}^{(0)}. \tag{9}
$$

Obviously, the original propagation probability matrix introduces high error, we improve it by scaling the aggregated messages and making it trainable. Formally, let $p_{ij} \in \mathbf{A}_{prop}$ correspond to the $i$-th row and $j$-th col of $\mathbf{A}_{prop}$, we define the scaling operator $\mathbf{d}_{ij} = \texttt{dis}(\mathbf{P}_{ii}, \mathbf{P}_{ij})$ for $j \neq i$, where $\texttt{dis}(\cdot)$ is a distance function or a function positively relative with the difference, which can be implemented using identity distance. Thus the corrected propagation matrix is expressed as

$$
\hat{\mathbf{A}}_{prop}^{(l)} = \mathbf{A}_{prop}^{(l)}/\mathbf{d}\mathbf{d}^{T} - \texttt{diag}(\mathbf{A}_{prop}^{(l)}). \tag{10}
$$

The purpose of it is to measure the global dependency of the current node through the probability difference. Then, we further model the positive and negative impacts of the messages to implement effective aggregation, which is formally represented as follows

$$
\mathbf{H}^{(l)} = \mathbf{W}\mathbf{H}^{(l-1)}, \ \mathbf{A}_{prop}^{(l)} = \hat{\mathbf{A}}_{prop}^{(l-1)} + \beta\left(\mathbf{H}^{(l)}\mathbf{H}^{(l)}\right)^{T},
$$
$$
\mathbf{H}_{pos}^{(l)} = \texttt{PoSign}(\hat{\mathbf{A}}_{prop}^{(l)})\mathbf{H}^{(l)}, \ \mathbf{H}_{neg}^{(l)} = \texttt{NeSign}(\hat{\mathbf{A}}_{prop}^{(l)})\mathbf{H}^{(l)}, \tag{11}
$$
$$
\mathbf{H}^{(l+1)} = \mathbf{H}^{(l)} + \mathbf{H}_{pos}^{(l)} + \mathbf{H}_{neg}^{(l)},
$$

where $\mathbf{H}^{(0)} = \mathbf{X}_{align}$, $\texttt{PoSign}(\cdot)$ and $\texttt{NeSign}(\cdot)$ represent the trainable adaptive propagation probabilities, it can be replaced by any reasonable nonlinear activation function. Here we analyze the error bound for the above adaptive heterogeneous propagation mechanism. The proof of the following theorem and reasonable assumptions are given in Appendix. A.1

**Theorem 3.1** *Suppose that the latent ground-truth mapping* $\Phi : \mathbf{x} \to y$ *from node features to node labels is differentiable and satisfies L-Lipschitz constraint, the following approximation error is*

$$
\left|\sum_{j \neq i}\mathbf{P}_{ij}^{\star}\Phi(\mathbf{H}^{(l)}) - \left(\mathbf{H}_{i}^{(l)} + \sum_{j \neq i}(\text{Pos}_{ij}^{(l)} + \text{Neg}_{ij}^{(l)})\mathbf{H}_{j}^{(l)}\right)\right|
$$
$$
\leq \left(L\left\|\epsilon_i\right\|_2 + \sum_{j \neq i}\mathbf{P}_{ij}^{\star}\ \mathcal{O}\left(\left\|\mathbf{H}_j^l - \mathbf{H}_i^{(l)}\right\|_2\right)\right) + \left(\left\|\mathbf{H}^{\star} - \phi\left(\kappa + \mathbf{P}\right)\mathbf{H}^{(l)}\right\|_2\right),
$$

*where* $\star$ *represents the global optimal,* $\epsilon$ *denotes immediate neighbors error,* $\mathcal{O}(\cdot)$ *denotes a higher order infinitesimal,* $\phi$ *and* $\kappa$ *represent propagation matrix and model differences, respectively.*

The core of the above propagation mechanisms is to generate embeddings based on other nodes in the embedding space. In other words, it means that any node representation can be mapped to a linear combination of existing node representations, which has been applied in many studies Zheng et al. (2022a); Yang et al. (2022). However, most of the methods use ranking mechanisms for representation and fail to consider modeling propagation processes, which has limitations.

## 4 Experiments

In this section, we conduct experimental analysis on five benchmark datasets with community split and structure Non-IID settings to validate the effectiveness of AdaFGL. We aim to answer the following five questions. **Q1**: Compared with other state-of-the-art FGL baselines, can AdaFGL achieve better predictive accuracy in the community split setting? **Q2**: How does structure Non-IID influence existing methods and can AdaFGL improve it? **Q3**: Are knowledge distillation and message detection working in adaptive propagation mechanisms? **Q4**: Why AdaFGL can achieve desirable predictions utilizing the interactions between decoupled modules? **Q5**: Compared with existing FGL methods and heterogeneous GNNs, what are the advantages of AdaFGL?

Table 3: The results of test accuracy based on Cora and Chameleon by implementing the community split: mean accuracy ± standard deviation. The best results are shown in bold.

| Community | Cora | | | Chameleon | | |
|---|---|---|---|---|---|---|
| | Client3 | Client5 | Client10 | Client3 | Client5 | Client10 |
| FedMLP | 63.65±1.03 | 64.91±1.33 | 72.44±1.26 | 44.88±0.97 | 42.46±1.42 | 42.88±1.40 |
| FedSGC | 81.71±0.06 | 81.58±0.12 | 82.07±0.23 | 33.57±0.23 | 30.44±0.21 | 30.03±0.19 |
| FedNLGNN | 82.16±1.01 | 82.47±0.88 | 84.04±1.31 | 55.92±1.92 | 53.94±1.54 | 39.31±3.18 |
| FedGGCN | 82.79±0.27 | 80.17±1.04 | 82.80±0.97 | 56.16±1.43 | 55.59±1.11 | 41.52±1.25 |
| FedGL | 81.47±0.76 | 82.88±1.09 | 83.01±1.61 | 48.13±1.23 | 42.62±2.06 | 39.95±1.21 |
| GCFL+ | 83.92±0.17 | 83.98±1.00 | 84.53±0.30 | 41.85±0.43 | 35.85±0.93 | 35.04±0.71 |
| FedSage+ | 82.80±0.63 | 83.06±0.55 | 85.49±0.91 | 54.16±1.67 | 53.08±1.79 | 50.11±1.95 |
| w/o HomoKD | 83.22±0.54 | 83.24±1.68 | 85.51±0.98 | 59.74±1.83 | 60.19±2.14 | 54.43±1.52 |
| w/o HeteTA | 84.91±0.57 | 85.87±1.75 | 86.47±0.83 | 57.98±1.27 | 58.96±1.35 | 51.28±1.06 |
| AdaFGL | **84.91±0.57** | **85.87±1.75** | **86.89±1.02** | **59.74±1.83** | **60.19±2.14** | **54.43±1.52** |

Table 4: The results of test accuracy are based on Cora and Chameleon by implementing the structure Non-IID: mean accuracy ± standard deviation. The best results are shown in bold.

| Non-IID | Cora | | | Chameleon | | |
|---|---|---|---|---|---|---|
| | Client3 | Client5 | Client10 | Client3 | Client5 | Client10 |
| FedMLP | 69.02±0.82 | 71.16±1.10 | 74.90±1.85 | 47.95±0.98 | 47.10±1.59 | 48.10±1.12 |
| FedSGC | 65.16±0.20 | 73.10±0.06 | 70.98±0.39 | 26.92±0.42 | 28.08±0.39 | 25.32±0.21 |
| FedNLGNN | 68.21±0.78 | 72.93±0.72 | 76.40±1.10 | 53.74±2.34 | 56.75±1.61 | 47.98±1.43 |
| FedGGCN | 75.19±0.50 | 78.20±1.04 | 78.78±0.98 | 55.59±0.84 | 60.51±1.53 | 49.90±1.17 |
| FedGL | 67.86±0.73 | 74.62±1.14 | 71.82±0.58 | 54.53±1.93 | 55.59±2.23 | 49.79±1.91 |
| GCFL+ | 68.81±0.36 | 74.15±0.64 | 72.05±0.39 | 41.15±0.83 | 44.64±0.66 | 45.18±0.61 |
| FedSage+ | 77.90±0.89 | 76.83±0.91 | 80.81±0.74 | 49.80±1.85 | 52.21±1.96 | 53.21±2.33 |
| w/o HomoKD | 78.34±0.43 | 80.54±0.97 | 80.38±1.28 | 60.89±1.12 | 67.85±1.83 | 59.89±2.37 |
| w/o HeteTA | 78.87±0.41 | 80.92±0.89 | 80.96±0.87 | 58.75±0.99 | 65.28±1.66 | 58.72±2.18 |
| AdaFGL | **79.03±0.46** | **81.15±1.18** | **82.32±1.34** | **61.05±1.16** | **68.41±1.97** | **60.28±2.49** |

## 4.1 Experimental Setup and Baselines

Existing FGL methods implement data partitioning by community split Wang et al. (2022b). We follow it while proposing a more convincing strategy structure Non-IID. Due to space limitations, the implementation details of the structure Non-IID can be found in Appendix. A.3 and Appendix. A.6. To demonstrate the effectiveness of AdaFGL, we combine powerful GNNs with FedAvg as the representative methods. Meanwhile, we compare the recently proposed FGL methods such as FedGL, FedSage+, and GCFL+. FedSGC efficiently exploits the local structure prior by performing feature propagation. FedNLGNN implements node embeddings by MLP or GCN to discover potential neighbors. FedGGCN further exploits the relationship between over-smoothing and heterogeneity to achieve weighted propagation. FedGL aims to optimize the local model performance using global information, it is essentially graph structure learning without overlapping nodes. FedSage+ performs local graph augmentation to improve prediction performance. GCFL+ implements the clustering process to perform the personalized update mechanisms. More details about baseline methods can be referred to Appendix. A.2.

## 4.2 Overall Performance

We first present the complete results on Cora and Chameleon in Table. 4 and Table. 3, which are two representative homogeneous and heterogeneous datasets. Due to the space limitation, the details about the experiment environment and results in other datasets can be found in AppendixA.6. Notably, since we randomly inject homogeneous or heterogeneous information into the structure Non-IID data partitioning process, the model performance does not directly relate to the number of clients. Meanwhile, in the community split setting, the process of model aggregation by multiple clients to achieve federated learning can be considered as ensemble learning. Therefore, the prediction performance gets better with the increasing number of clients in some cases.

To answer **Q1**, Table. 3 shows the comparison results with the baseline methods in community split setting. For the homogeneous dataset Cora, compared with the most competitive FGL methods, AdaFGL achieved accuracy gains of 1.18%, 2.25%, and 1.64% in multiple client settings, respec-

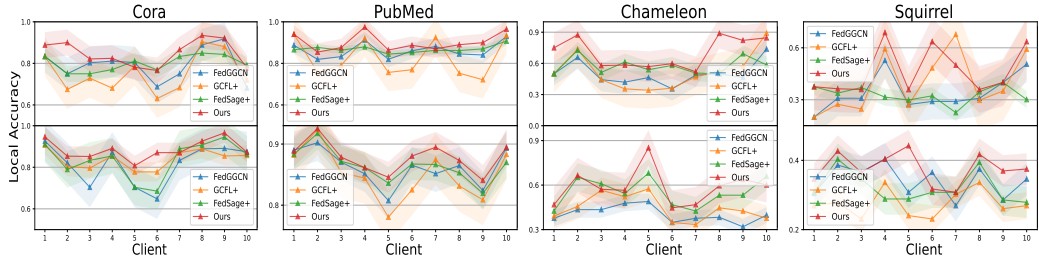

Figure 4: The results based on two data partitioning methods on 10 clients, where the upper part represents structure Non-IID and the lower part represents community split.

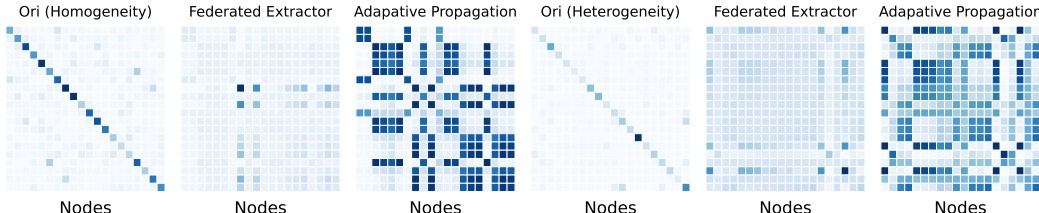

Figure 5: The heat maps of nodes similarity on PubMed for 10 clients. Homogeneity and heterogeneity represent local structure assumptions, respectively. Dark color indicates high similarity.

tively. Meanwhile, AdaFGL exceeds the best methods among all considered baselines on the heterogeneous datasets by a margin of $6.37\%$ to $8.52\%$. In the community split setting, we improve the prediction accuracy by utilizing the local smoothing prior and adaptive propagation mechanisms.

To answer **Q2**, We demonstrate the performance of existing methods in the face of structure Non-IID challenges in Table. 4. Although FedGGCN performs well in general, it cannot obtain competitive performance. Despite FedSage+ achieving effective local graph augmentation by sharing global data, structure Non-IID is a natural challenge, and this weakness is amplified when heterogeneity is high. In contrast, our method achieves performance gains of $1.45\%$, $3.77\%$, and $1.87\%$ compared to the highest prediction accuracy. Impressively, AdaFGL improves performance by $9.82\%$, $13.06\%$, and $13.29\%$ in the structure Non-IID setting for the heterogeneous dataset Chameleon. From the observation of the comparison results with the baselines, our method has significant advantages, especially in terms of robustness and impressive performance.

### 4.3 ABLATION EXPERIMENTS

To answer **Q3**, we present the ablation experiment results in Table. 3 and Table. 4, where HomoKD represents the online distillation in the homogeneous propagation module and HeteTA represents the trainable probability propagation matrix in the heterogeneous propagation module. We observe that the online distillation enhances Homogeneous propagation by combining local smoothing features and local embeddings, it can effectively improve model performance without adding additional computation costs. In essence, it achieves mutually supervised end-to-end learning of global and local information. Furthermore, the trainable probability propagation matrix optimizes the heterogeneity propagation module. It learns the global optimal propagation mechanism and detects positive and negative messages to generate embeddings. HeteTA can discover the global dependence of the current node and achieve effective message aggregation, which is proved by Theorem. 3.1.

### 4.4 VISUALIZATION AND EXPLAINABILITY ANALYSIS

To answer **Q4**, we present the local prediction accuracy trends with the competitive baseline methods in Fig. 4. According to it, we can notice that our method achieves the best performance in most cases under both community split and structure Non-IID data settings, while the overall trend is optimized. Due to space limitations, the relevant experimental results about the hyperparameter sensitivity analysis experiments on AdaFGL and conclusions can be found in Appendix. A.5.

In order to illustrate the effectiveness of the federated global knowledge extractor and the adaptive propagation mechanisms, we also analyze the explainability by presenting the heat maps shown

Table 5: A summary of very recent FGL methods and our approach.

| Method | Type | Exchange Messages | Structure Non-IID |
|---|---|---|---|
| FedSage+ | Augmentation | Model Params (GraphSAGE, NeighGen), Node Embeddings | ✗ |
| FedGL | Augmentation | Model Params (Linear Regression or MLP) | ✗ |
| GCFL+ | Personalization | Model Params (Linear Regression or MLP), Model Gradient | ✗ |
| Ours | Personalization | Model Params (Linear Regression or MLP) | ✓ |

Table 6: A summary of powerful GNNs in heterogeneous graph and our approach.

| Method | Neighbor Discovery | Message Combination | Strategy | Heterogeneity |
|---|---|---|---|---|
| MLP | ✗ | ✗ | Ignore Structure, Update Function | ✗ |
| FedGL | ✓ | ✗ | Graph Structure Learning, Update Function | ✗ |
| FedSage+ | ✓ | ✗ | Graph Augmentation, Update Function | ✗ |
| NLGNN | ✓ | ✗ | Embedding Model, Similarity Ranking | ✓ |
| GGCN | ✗ | ✓ | Nodal Degree Weighting, Update Function | ✓ |
| Ours | ✓ | ✓ | Trainable Propagation Matrix | ✓ |

in Fig. 5. We perform structure Non-IID partitioning for 10 clients on PubMed, then select the client with the highest number of nodes with homogeneity and heterogeneity. Based on this, we randomly sampled 20 nodes to obtain the similarity score by computing the embedding transpose. From the observation of the results, we notice that the federated global knowledge extractor only obtains fuzzy results and cannot be optimized for the local subgraphs. Fortunately, we achieve an effective combination of global knowledge and local subgraph structure prior to obtaining explicit node embeddings, which is also demonstrated through the final output in Fig. 5.

## 4.5 METHODS COMPARISON

To answer **Q5**, we review three recent FGL methods and analyze our approach to them in terms of three aspects: method type, exchange messages, and the ability to solve structure Non-IID problems as shown in Table.5. Obviously, although FedSage+ can achieve competitive results, it introduces significant communication costs and privacy concerns. Specifically, FedSage+ trains two models and thus has communication costs, while implementing cross-client information sharing to improve predictive performance, which no doubt increases privacy concerns. GCFL+ has limitations in model selection leads to its failure to handle the structure Non-IID problem in subgraph learning. In our experiments, FedGL is essentially a local graph structure learning process. In contrast, our approach can utilize the computational capabilities of the local system while minimizing communication costs and privacy concerns. More experimental details can be found in Appendix. A.4.

Then, we compare the effectiveness of existing GNNs and our approach to handling heterogeneous graph, which focuses on two points: Neighbor Discovery and Message Combination, which is shown in Fig. 6. We observe that MLP ignores graph structure prior which leads to the failure to handle heterogeneous graphs. Although FedGL and FedSage+ can improve this problem by utilizing global information for local graph augmentation, the limitations of propagation lead to the fact that they are still not the best solutions. Notably, they cannot handle the structure Non-IID problem in FGL. Although NLGNN and GGCN attempt to solve the heterogeneous structure problem, they cannot be directly applied in FGL. Therefore, we are motivated by these methods and propose adaptive propagation mechanisms to improve the performance, which has been validated to be effective.

## 5 CONCLUSION

In this paper, we discover and define the structure Non-IID problem in FGL, which is a new challenge for existing methods. Based on this, we propose a new paradigm AdaFGL for more general federated data settings. Specifically, we investigate the structure Non-IID problem in FGL for supplementing the existing community split data partitioning approach, which is a more practical federated data setting. To implement effective FGL on heterogeneous distributed subgraphs, we propose AdaFGL which consists of the federated global knowledge extractor and adaptive propagation modules. It combines FL and GNNs tightly and benefits from their evolution. Extensive experiments based on the community split and structure Non-IID data settings demonstrate the effectiveness of AdaGFL. We believe that the ability to fully utilize the graph structure information is the key to achieving efficient FGL, thus the research on graph structure in FGL is a promising direction.

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

## A   Appendix Outline

The appendix is organized as follows:

**A.1** Theory error bounds for adaptive heterogeneous propagation modules.

**A.2** More details about the compared baselines.

**A.3** Datasets description and structure Non-IID data setting.

**A.4** Communication costs analysis.

**A.5** Hyperparameter sensitivity analysis.

**A.6** Experimental environment and additional base results.

### A.1   Theory error bounds for adaptive heterogeneous propagation

To demonstrate the effectiveness of the adaptive heterogeneous propagation module, we prove its error bound. We first make the reasonable following assumption and definitions.

**Assumption A.1**  $\Phi$ *is L-smooth,* $\forall \mathbf{x}_1, \mathbf{x}_2 \in \mathtt{dom}(\Phi)$

$$\Phi(\mathbf{x}_1) \leq \Phi(\mathbf{x}_2) + (\mathbf{x}_1 - \mathbf{x}_2)^T \nabla \Phi(\mathbf{x}_2) + \frac{L}{2} ||\mathbf{x}_1 - \mathbf{x}_2||_2^2.$$

Then we quote the embedding method theorem Linial et al. (1995).

**Definition A.1** *Given two metric spaces* $(\mathcal{V}, d)$ *and* $(Z, d')$ *and mapping function* $\Phi : \mathcal{V} \rightarrow \mathcal{Z}$*, the distortion* $\epsilon_{distor}$ *is definied as* $\forall u, v \in \mathcal{V}, 1/\epsilon_{distor} d(u, v) \leq d'(\Phi(u), \Phi(v)) \leq d(u, v)$.

**Theorem A.1** *(Bourgain theorem) Given any finite metric space* $(\mathcal{V}, d)$ *with* $\mathcal{V} = n$*, there exists an embedding of* $(\mathcal{V}, d)$ *into* $\mathbb{R}^k$ *under any* $l_p$ *metric, where* $k = O(\log^2 n)$*, and the distortion of the embedding is* $O(\log n)$.

It defines the distortion $O(\log n)$ in the embedding space $(\mathcal{V}, d)$ for mapping methods satisfying the above conditions. Based on this, we consider a graph $G$ with fixed structure represented by $\tilde{\mathbf{A}} = \hat{\mathbf{D}}^{-1/2} \hat{\mathbf{A}} \hat{\mathbf{D}}^{-1/2}$, embeddings represented with $\mathbf{H}$ in the forward propagation, and nodes mapping function $\Phi(\mathbf{H})$, which satisfies the Theorem. A.1, it can be expressed as

$$\phi(\mathbf{H}) = \left( \frac{d(\mathbf{H}, S_{1,1})}{k}, \frac{d(\mathbf{H}, S_{1,2})}{k}, \dots, \frac{d(\mathbf{H}, S_{\log n, c \log n})}{k} \right),$$

where $d(\mathbf{H}, S_{i,j}) = \min_{u \in S_{i,j}} d(\mathbf{H}, u)$. $S_{i,j} \subset \mathcal{V}, i = 1, 2, \dots, \log n, j = 1, 2, \dots, c \log n$ represents $c \log^2 n$ random sets, where $c$ is a constant. It is chosen by including each point in $\mathcal{V}$ independently with probability $1/2^i$. Then motivated by Xie et al. (2021) and the above conclusions, we have the following model weights difference proposition.

**Proposition A.1** *Assume the propagation probability matrix, hidden embeddings, and label difference with global optima* $f_\theta^\star$ *and local model* $f_\theta$ *are bounded with*

$$\|\mathbf{P}^\star - \mathbf{P}\|_2^2 = \|E_{\mathbf{P}}\|_2^2 \leq \epsilon_{\mathbf{P}}$$
$$\|\mathbf{H}^\star - \mathbf{H}\|_2^2 = \|E_{\mathbf{H}}\|_2^2 \leq \epsilon_{\mathbf{H}}$$
$$\left\|\hat{\mathbf{Y}}^\star - \hat{\mathbf{Y}}\right\|_2^2 = \left\|E_{\hat{\mathbf{Y}}}\right\|_2^2 \leq \epsilon_{\hat{\mathbf{Y}}}.$$

Based on this, given that $\|\mathbf{H} \cdot \mathbf{H}^\star\|_2^2 = \|\mathbf{H} \cdot (\mathbf{H} + E_\mathbf{H})\|_2^2 \geq \|\mathbf{H}E_\mathbf{H}\|_2^2$. Let $\|\mathbf{X}E_\mathbf{H}\|_2^2 = \delta_\mathbf{H}$, then we have $\left\|\mathbf{H}^{\star-1} - \mathbf{H}^{-1}\right\|_2^2 = \|E_{\mathbf{H}^{-1}}\| \leq \epsilon_\mathbf{H}/\delta_\mathbf{H}$. If we choose SGC Wu et al. (2019) for the forward propagation, the model weights difference with the influence of feature difference is represented as

$$\phi = \|f_\theta^\star - f_\theta\|_2 = \left\|(\mathbf{P}\mathbf{H}^\star)^{-1}\hat{\mathbf{Y}}^\star - (\mathbf{P}\mathbf{H})^{-1}\hat{\mathbf{Y}}\right\|_2^2$$

$$= \left\|\mathbf{H}^{\star-1}\mathbf{P}^{-1}(\hat{\mathbf{Y}} + E_{\hat{\mathbf{Y}}}) - \mathbf{H}^{-1}\mathbf{P}^{-1}\hat{\mathbf{Y}}\right\|_2^2$$

$$= \left\|(\mathbf{H}^{\star-1} - \mathbf{H}^{-1})\mathbf{P}^{-1}\hat{\mathbf{Y}} + \mathbf{H}^{\star-1}\mathbf{P}^{-1}E_{\hat{\mathbf{Y}}}\right\|_2^2$$

$$= \left\|E_{\mathbf{H}^{-1}}\mathbf{P}^{-1}\hat{\mathbf{Y}} + (\mathbf{P}\mathbf{H} + \mathbf{P}E_\mathbf{H})^{-1}E_{\hat{\mathbf{Y}}}\right\|_2^2$$

$$\leq \frac{\epsilon_\mathbf{H}}{\delta_\mathbf{H}}\left\|\mathbf{P}^{-1}\hat{\mathbf{Y}}\right\|_2^2 + \frac{\epsilon_\mathbf{H}^2\epsilon_{\hat{\mathbf{Y}}}}{\delta_\mathbf{X}}\left\|(\mathbf{P}\mathbf{H})^{-1}\right\|_2^2 + \epsilon_\mathbf{H}\epsilon_{\hat{\mathbf{Y}}}\left\|(\mathbf{P}\mathbf{H})^{-1}\right\|_2^4.$$

Similarly, there exists $\|\mathbf{P} \cdot \mathbf{P}^\star\|_2^2 = \|\mathbf{P} \cdot (\mathbf{P} + E_\mathbf{P})\|_2^2 \geq \|\mathbf{P}E_\mathbf{P}\|_2^2$, $\|\mathbf{P}E_\mathbf{P}\|_2^2 = \delta_\mathbf{P}$, and $\left\|\mathbf{P}^{\star-1} - \mathbf{P}^{-1}\right\|_2^2 = \|E_{\mathbf{P}^{-1}}\| \leq \epsilon_\mathbf{P}/\delta_\mathbf{P}$.

we can obtain the model weight differences with the influence of structure difference.

$$\phi = \|f_\theta^\star - f_\theta\|_2 = \left\|(\mathbf{P}^\star\mathbf{H})^{-1}\hat{\mathbf{Y}}^\star - (\mathbf{P}\mathbf{H})^{-1}\hat{\mathbf{Y}}\right\|_2^2$$

$$= \left\|\mathbf{H}^{-1}\left(\mathbf{P}^{\star-1}\hat{\mathbf{Y}}^\star - \mathbf{P}^{-1}\hat{\mathbf{Y}}\right)\right\|$$

$$= \left\|\mathbf{H}^{-1}\right\|_2^2\left\|(\mathbf{P}^{-1} + E_{\mathbf{P}^{-1}})(\hat{\mathbf{Y}} + E_{\hat{\mathbf{Y}}}) - \mathbf{P}^{-1}\hat{\mathbf{Y}}\right\|_2^2$$

$$= \left\|\mathbf{H}^{-1}\right\|_2^2\left\|\mathbf{P}^{-1}E_{\hat{\mathbf{Y}}} + E_{\mathbf{P}^{-1}}\hat{\mathbf{Y}} + E_{\mathbf{P}^{-1}}E_{\hat{\mathbf{Y}}}\right\|$$

$$\leq \left\|\mathbf{H}^{-1}\right\|_2^2\left[\epsilon_{\hat{\mathbf{Y}}}\left\|\mathbf{P}^{-1}\right\|_2^2 + \frac{\epsilon_\mathbf{P}}{\delta_\mathbf{P}}\left\|\hat{\mathbf{Y}}\right\|_2^2 + \frac{\epsilon_\mathbf{P}\epsilon_{\hat{\mathbf{Y}}}}{\delta_\mathbf{P}}\right].$$

**Proof A.1** *Here, based on the Eq. 11, we consider the adaptive heterogeneous propagation process*

$$\mathbf{H}^{(l+1)} = \mathbf{H}^{(l)} + \mathbf{H}_{pos}^{(l)} + \mathbf{H}_{neg}^{(l)}$$

$$= \mathbf{H}^{(l)} + \text{PoSign}(\hat{\mathbf{A}}_{prop}^{(l)})\mathbf{H}^{(l)} + \text{NeSign}(\hat{\mathbf{A}}_{prop}^{(l)})\mathbf{H}^{(l)}$$

$$= \mathbf{H}^{(l)} + \text{PoSign}\left(\hat{\mathbf{A}}_{prop}^{(l-1)} + \beta\mathbf{W}\mathbf{H}^{(l-1)}(\mathbf{W}\mathbf{H}^{(l-1)})^T\right)\mathbf{H}^{(l)}$$

$$+ \text{NeSign}\left(\hat{\mathbf{A}}_{prop}^{(l-1)} + \beta\mathbf{W}\mathbf{H}^{(l-1)}(\mathbf{W}\mathbf{H}^{(l-1)})^T\right)\mathbf{H}^{(l)}.$$

*Take node $i$ as an example, given that $\Phi(\cdot)$ is differentiable, where contains the gradient update of the model difference $\phi$. Meanwhile, in order to quantify the difference between our trainable propagation probability matrix and the global optimum, we define*

$$\kappa_i = \sum \mathbf{P}^\star[i :] - \left(\hat{\mathbf{A}}_{prop}^0[i :] + \sum_l \beta\mathbf{W}\mathbf{H}^l(\mathbf{W}\mathbf{H}^{(l)})^T[i :]\right),$$

*where $\mathbf{P}^\star$ represents the optimal propagation probability matrix. Then, we use $\text{Pos}, \text{Neg}$ to denote the positive and negative message propagation weights $\mathbf{P} = \hat{\mathbf{A}}_{prop}^{(l)}$, there exist*

$$\mathbf{H}_i^{(l+1)} = \sum_{j \neq i}\mathbf{P}_{ij}^\star\Phi(\mathbf{H}^{(l)})$$

$$= \mathbf{H}_i^{(l)} + \sum_{j \neq i}(\text{Pos}_{ij}^{(l)} + \text{Neg}_{ij}^{(l)})\mathbf{H}_j^{(l)}$$

$$= \left(\kappa_i + \mathbf{P}_{ii} + \sum_{j \neq i}(\text{Pos}_{ij}^{(l)} + \text{Neg}_{ij}^{(l)})\right)\phi\mathbf{H}^{(l)},$$

*where* $\text{Pos}_{ij}^{(l)} + \text{Neg}_{ij}^{(l)} = \hat{\mathbf{A}}_{prop}^{(l)}[i :]$. *Then, we perform a first-order Taylor expansion with Peano's form of remainder at* $\mathbf{H}_i^{(l)}$ *and consider the model differences*

$$\sum_{j \neq i} \mathbf{P}_{ij}^{\star} \Phi(\mathbf{H}^{(l)}) = \sum_{j \neq i} \mathbf{P}_{ij}^{\star} \left( \Phi(\mathbf{H}^{(l)}) + \frac{\partial \Phi(\mathbf{H}_j^{(l)})}{\partial (\mathbf{H}^{(l)})^T} (\mathbf{H}_j^{(l)} - \mathbf{H}_i^{(l)}) + \mathcal{O}(\|\mathbf{H}_j^{(l)} - \mathbf{H}_i^{(l)}\|_2) \right)$$

$$= \sum_{j \neq i} \mathbf{P}_{ij}^{\star} \Phi(\mathbf{H}^{(l)}) + \sum_{j \neq i} \mathbf{P}_{ij}^{\star} \frac{\partial \Phi(\mathbf{H}_j^{(l)})}{\partial (\mathbf{H}^{(l)})^T} (\mathbf{H}_j^{(l)} - \mathbf{H}_i^{(l)}) + \sum_{j \neq i} \mathbf{P}_{ij}^{\star} \mathcal{O}(\|\mathbf{H}_j^{(l)} - \mathbf{H}_i^{(l)}\|_2).$$

*Now, we let* $\sum_{j \neq i} \mathbf{P}_{ij}^{\star} (\mathbf{H}_j^{(l)} - \mathbf{H}_i^{(l)}) = -\epsilon_i$, *there exist*

$$\sum_{j \neq i} \mathbf{P}_{ij}^{\star} \Phi(\mathbf{H}^{(l)}) = \left( \kappa_i + \mathbf{P}_{ii} + \sum_{j \neq i} (\text{Pos}_{ij}^{(l)} + \text{Neg}_{ij}^{(l)}) \right) \phi \mathbf{H}^{(l)},$$

$$= \sum_{j \neq i} \mathbf{P}_{ij}^{\star} \Phi(\mathbf{H}^{(l)}) - \frac{\partial \Phi(\mathbf{H}_j^{(l)})}{\partial (\mathbf{H}^{(l)})^T} \epsilon_i + \sum_{j \neq i} \mathbf{P}_{ij}^{\star} \mathcal{O}(\|\mathbf{H}_j^{(l)} - \mathbf{H}_i^{(l)}\|_2)$$

$$\left| \sum_{j \neq i} \mathbf{P}_{ij}^{\star} \Phi(\mathbf{H}^{(l)}) - \left( \kappa_i + \mathbf{P}_{ii} + \sum_{j \neq i} (\text{Pos}_{ij}^{(l)} + \text{Neg}_{ij}^{(l)}) \right) \phi \mathbf{H}^{(l)} \right| = \left| \frac{\partial \Phi(\mathbf{H}_j^{(l)})}{\partial (\mathbf{H}^{(l)})^T} \epsilon_i - \sum_{j \neq i} \mathbf{P}_{ij}^{\star} \mathcal{O}(\|\mathbf{H}_j^{(l)} - \mathbf{H}_i^{(l)}\|_2) \right|.$$

*According to Cauchy-Schwarz inequality and L-Lipschitz property, we have*

$$\left| \frac{\partial \Phi(\mathbf{H}_i^{(l)})}{\partial (\mathbf{H}^{(l)})^T} \epsilon_i \right| \leq \left\| \frac{\partial \Phi(\mathbf{H}_i^{(l)})}{\partial (\mathbf{H}^{(l)})^T} \right\| \|\epsilon_i\|_2 \leq L \|\epsilon_i\|_2.$$

*Therefore, the approximation of* $\mathbf{H}_i^{(l)} + \sum_{j \neq i} (\text{Pos}_{ij}^{(l)} + \text{Neg}_{ij}^{(l)}) \mathbf{H}_j^{(l)}$ *is bounded by*

$$\left| \sum_{j \neq i} \mathbf{P}_{ij}^{\star} \Phi(\mathbf{H}^{(l)}) - \left( \mathbf{H}_i^{(l)} + \sum_{j \neq i} (\text{Pos}_{ij}^{(l)} + \text{Neg}_{ij}^{(l)}) \mathbf{H}_j^{(l)} \right) \right|$$

$$= \left| \frac{\partial \Phi(\mathbf{H}_i^{(l)})}{\partial (\mathbf{H}^{(l)})^T} - \sum_{j \neq i} \mathbf{P}_{ij}^{\star} \mathcal{O} \left( \left\| \mathbf{H}_j^l - \mathbf{H}_i^{(l)} \right\|_2 \right) \right| + \left( \left\| \mathbf{H}^{\star} - \mathbf{P}^{\star} \mathbf{H}^{(l)} \right\|_2 \right)$$

$$\leq \left| \frac{\partial \Phi(\mathbf{H}_i^{(l)})}{\partial (\mathbf{H}^{(l)})^T} \epsilon_i \right| + \left| \sum_{j \neq i} \mathbf{P}_{ij}^{\star} \mathcal{O} \left( \left\| \mathbf{H}_j^l - \mathbf{H}_i^{(l)} \right\|_2 \right) \right| + \left( \left\| \mathbf{H}^{\star} - \mathbf{P}^{\star} \mathbf{H}^{(l)} \right\|_2 \right)$$

$$\leq \left( L \|\epsilon_i\|_2 + \sum_{j \neq i} \mathbf{P}_{ij}^{\star} \mathcal{O} \left( \left\| \mathbf{H}_j^l - \mathbf{H}_i^{(l)} \right\|_2 \right) \right) + \left( \left\| \mathbf{H}^{\star} - \phi \left( \kappa + \mathbf{P} \right) \mathbf{H}^{(l)} \right\|_2 \right),$$

*where* $\mathbf{H}^{\star}$ *represents the global optimal embeddings. Based on this, we obtain the theory error bound for heterogeneous propagation. From the observation of error bounds, we reveal that in theory, the adaptive heterogeneous propagation process can minimize the immediate neighbors error* $\epsilon_i$, *the model difference* $\phi$, *and the propagation probability matrix difference* $\kappa$ *to scale the error to improve the predictive performance.*

## A.2 COMPARED BASELINES

The main characteristic of all baselines are listed below:

FedMLP: The combination of FedAvg and MLP, we employ a two-layer MLP with the hidden dimension of 64. It generates node embeddings based on the original features while ignoring graph structure information in the forward propagation process.

Table 7: Statistics of five benchmark datasets.

| Datasets | #Nodes | #Edges | #Classes | Nodes Homo | Edges Homo |
|----------|--------|--------|----------|------------|------------|
| Cora | 2708 | 5278 | 7 | 0.83 | 0.81 |
| CiteSeer | 3703 | 4552 | 6 | 0.71 | 0.74 |
| PubMed | 19717 | 44324 | 3 | 0.79 | 0.80 |
| Chameleon | 2277 | 36101 | 5 | 0.25 | 0.24 |
| Squirrel | 5201 | 217073 | 5 | 0.22 | 0.22 |

FedSGC: We combine FedAvg and SGC Wu et al. (2019), we default to use the 3-layer feature propagation process, which follows the homogeneous assumption and thus fails to deal with heterogeneous graph.

FedNLGNN: Implementation of NLGNN (NLMLP or NLGCN) Liu et al. (2021) based on FedAvg, we select the more effective version to present the model performance. It depends on the embedding model and suffers from representational limitations.

FedGGCN: The combination of FedAvg and GGCN Yan et al. (2021), we follow the experimental setup of the original paper as much as possible, which can handle heterogeneous graphs effectively, but cannot achieve competitive results on homogeneous graphs.

FedGL Chen et al. (2021): As a FGL training framework, it strongly relies on the overlapping nodes assumption, which in our data setting is essentially local graph structure learning.

GCFL+ Xie et al. (2021): due to the limitations of personalized techniques in model selection, they cannot fundamentally solve the structure Non-IID challenges.

FedSage+ Zhang et al. (2021): It trains NeighGen to achieve local subgraph augmentation by sharing global missing subgraph feature and topology information for the most powerful results, but suffers from privacy leakage risk and additional computational costs.

For fairness, we follow the experimental setup of the baseline methods paper as much as possible, and in other cases, we show the best prediction accuracy. In addition, the number of rounds for the above baseline methods is 50, and the local epoch is 20.

### A.3 DATASETS DESCRIPTION AND STRUCTURE NON-IID DATA SETTING

The statistics of datasets are summarized in Table. 7, which contains both homogeneity and heterogeneity. In our experiments, we use five benchmark datasets containing homogeneity and heterogeneity, for which details are given below.

**Cora**, **Citeseer**, and **Pubmed** Yang et al. (2016b) are three popular citation network datasets. In these three networks, papers from different topics are considered as nodes, and the edges are citations among the papers. The node attributes are binary word vectors, and class labels are the topics papers belong to.

**Chameleon** and **Squirrel** are two web page datasets collected from Wikipedia Rozemberczki et al. (2021), where nodes are web pages on specific topics and edges are hyperlinks between them.

Based on this, we illustrate the structure Non-IID data partitioning process in detail. The core of it is the Dirichlet process He et al. (2020), Its basic analysis is as follows. The pdf of the Dirichlet distribution is defined as

$$p(P = \{p_i\}|\alpha_i) = \frac{\prod_i \Gamma(\alpha_i)}{\Gamma(\sum_i \alpha_i)} \Gamma_i p_i^{\alpha_i - 1},$$

where $\alpha_i \in \{\alpha_1, \ldots, \alpha_k\} > 0$ is the dimensionless distribution parameter, the scale (or concentration) $\vartheta = \sum_i \alpha_i$, the base measure $(\alpha_1^\star, \ldots, \alpha_k^\star), \alpha_i^\star = \alpha_i/\vartheta$, and $\Gamma(n) = (n-1)!$. Dirichlet is a distribution over Multinomials, thus there is $\sum_i p_i = 1, p_i \geq 0$, where $p_i$ represents the probability.

Table 8: The results of accuracy based on Cora and CiteSeer by implementing the community split of 10 clients: mean accuracy $\pm$ standard deviation. The best results are shown in bold.

| Dataset | Cora | | | CiteSeer | | |
|---|---|---|---|---|---|---|
| Method | # Val Acc | # Test Acc | # Params | # Val Acc | # Test Acc | # Params |
| FedSage | 85.32±0.87 | 85.49±0.91 | $1.52 \times 10^7$ | 76.13±0.54 | 75.49±0.63 | $3.85 \times 10^7$ |
| FedGGCN | 82.34±0.84 | 82.80±0.97 | $5.52 \times 10^5$ | 75.32±0.95 | 75.78±1.12 | $1.42 \times 10^6$ |
| Ours | **87.97±0.98** | **86.89±1.02** | $\mathbf{2.88 \times 10^5}$ | **79.63±1.48** | **77.05±1.72** | $\mathbf{7.26 \times 10^5}$ |

Table 9: The results of accuracy based on Chameleon and Squirrel by implementing the community split of 10 clients: mean accuracy $\pm$ standard deviation. The best results are shown in bold.

| Dataset | Chameleon | | | Squirrel | | |
|---|---|---|---|---|---|---|
| Method | # Val Acc | # Test Acc | # Params | # Val Acc | # Test Acc | # Params |
| FedSage | 53.49±2.06 | 50.11±1.95 | $2.44 \times 10^7$ | 31.44±2.17 | 31.27±1.93 | $2.19 \times 10^7$ |
| FedGGCN | 43.81±1.37 | 41.52±1.25 | $8.95 \times 10^5$ | 35.26±1.10 | 34.10±1.25 | $8.04 \times 10^5$ |
| Ours | **58.06±1.23** | **54.43±1.52** | $\mathbf{8.95 \times 10^5}$ | **39.16±1.88** | **35.43±2.53** | $\mathbf{8.04 \times 10^5}$ |

It determines the mean distribution, and the scale affects the variance, then we obtain

$$E(p_i) = \frac{\alpha_i}{\vartheta} = \alpha_i^\star,$$

$$Var(p_i) = \frac{\alpha_i(\vartheta - \alpha)}{\vartheta^2(\vartheta + 1)} = \frac{\alpha_i^\star(1 - \alpha_i^\star)}{(\vartheta + 1)},$$

$$Cov(p_i, p_j) = \frac{-\alpha_i \alpha_j}{\vartheta^2(\vartheta + 1)},$$

which means that a Dirichlet with small scale $\vartheta$ favors extreme distributions, but this prior belief is very weak and is easily overwritten by data. As $\vartheta \to \infty$, the covariance $\to 0$, the samples $\to$ base measure.

Based on this, we start sampling the edges to determine the attribution of a pair of nodes. If a conflicting set of nodes exists it is sampled again and finally generates induced subgraphs. Then, we randomly inject homogeneous or heterogeneous information based on the label prior, which can solve unreal structure loss and enhance structure identity. We propose to set three probabilities $p_{iso}$, $p_{homo}$, and $p_{hete}$ for each client individually to represent the probability of avoiding isolated nodes, increasing homogeneous edges, and increasing heterogeneous edges in the subgraph, respectively. Specifically, $p_{iso}$ represents the probability of isolated nodes generating edges with other nodes, which can effectively be used to prevent the generation of isolated nodes. $p_{homo}$ applies to the subgraph of clients that are selected to enhance homogeneity, and it represents the probability of connection between two nodes with the same label based on the label prior information. Correspondingly, $p_{hete}$ represents the probability used to perform the structure information injection for the client subgraph that performs the heterogeneity enhancement.

## A.4 COMMUNICATION COSTS ANALYSIS

The advantage of our approach is to exploit the local structure prior while making full use of the global information, which considers the characteristics of GNNs. It has the benefit of reducing the communication costs and privacy concerns during the federated training process. Meanwhile, thanks to the utilization of the local structure information, we can obtain models with better representational power to improve the performance. To demonstrate the effectiveness of our method, we provide the experimental results of AdaFGL with the two most competitive methods, FedSage and FedGGCN, as shown in Tables. A.4 and Table. A.4.

According to the experimental results, we observe that AdaFGL maintains the low communication costs and achieves a satisfying result, which mainly benefits from the utilization of local structure information by the adaptive propagation modules. Compared to FedSage, which is the current most competitive FGL approach, suffers from the performance improvement and communication costs dilemma, which also brings more privacy concerns.

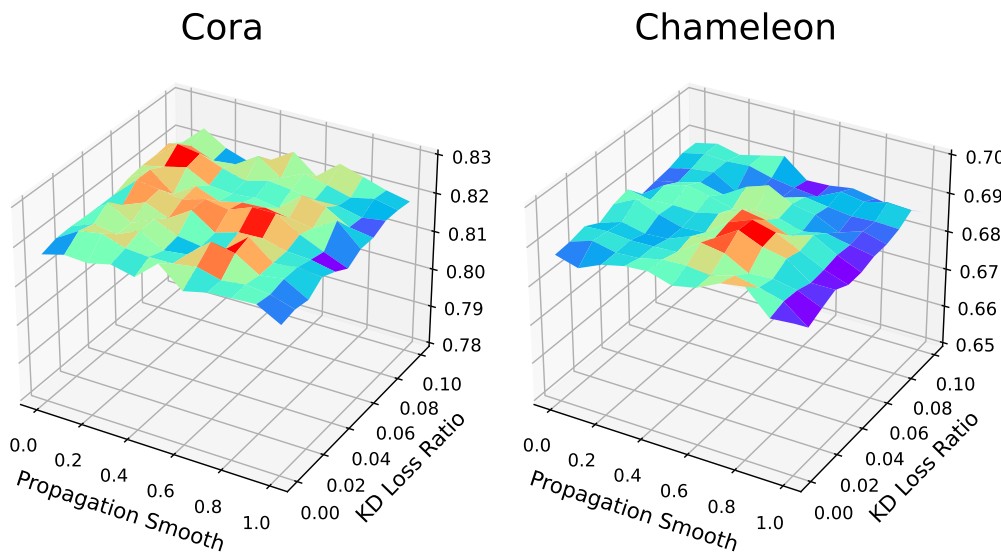

Figure 6: Hyperparameter sensitivity analysis on the Cora and Chameleon for 5 clients.

Table 10: The results of test accuracy based on CiteSeer and Squirrel by implementing the community split: mean accuracy ± standard deviation. The best results are shown in bold.

| | CiteSeer | | | Squirrel | | |
|---|---|---|---|---|---|---|
| Community | Client3 | Client5 | Client10 | Client3 | Client5 | Client10 |
| FedMLP | 65.83±0.66 | 70.54±1.42 | 70.86±2.03 | 29.90±0.59 | 40.89±0.64 | 30.58±0.70 |
| FedSGC | 72.59±0.40 | 73.31±0.05 | 75.68±0.10 | 29.98±0.13 | 32.59±1.06 | 27.96±0.59 |
| FedNLGNN | 71.35±0.78 | 73.14±0.89 | 75.05±0.65 | 40.79±1.58 | 44.32±1.66 | 32.22±1.24 |
| FedGGCN | 70.02±0.57 | 73.30±0.70 | 75.78±1.12 | 42.76±0.57 | 48.45±1.03 | 34.10±1.25 |
| FedGL | 71.57±1.05 | 72.28±0.62 | 74.40±0.78 | 32.45±2.74 | 35.32±1.04 | 29.96±0.56 |
| GCFL+ | 73.25±0.27 | 73.30±0.49 | 76.28±0.26 | 27.17±0.46 | 28.03±0.30 | 27.33±0.39 |
| FedSage+ | 69.15±0.62 | 71.88±0.73 | 75.49±0.63 | 36.72±1.05 | 43.65±1.46 | 31.27±1.93 |
| w/o HomoKD | 71.46±0.71 | 74.26±0.88 | 76.12±1.59 | 43.02±1.19 | 49.38±1.62 | 35.43±2.53 |
| w/o HeteTA | 72.57±0.50 | 73.57±0.39 | 75.81±1.06 | 41.94±0.71 | 47.79±1.10 | 33.16±1.40 |
| AdaFGL | **73.69±0.65** | **75.32±0.90** | **77.05±1.72** | **43.02±1.19** | **49.38±1.62** | **35.43±2.53** |

## A.5 HYPERPARAMETER SENSITIVITY ANALYSIS

Here we conduct the hyperparameter sensitivity of AdaFGL, and the experimental results are shown in Fig. 6. In our experiments, we analyze the ratio of online distillation loss in the homogeneous propagation module and the smoothing coefficient of the trainable propagation matrix in the heterogeneous propagation module. According to the experimental results, we observe that AdaFGL performs robustness except for extreme cases. Furthermore, we obtain the conclusion from the results generated by the extreme knowledge distillation loss ratios, where the low confidence base predictor results instead affect the homogeneous propagation module. Motivated by this, in order to avoid global embeddings with low confidence from influencing the propagation module, we measure the confidence of the global model according to the characteristics of the base predictor.

## A.6 EXPERIMENTAL ENVIRONMENT AND ADDITIONAL BASE RESULTS

The experiments are conducted on a machine with Intel(R) Xeon(R) Gold 6230R CPU @ 2.10GHz, and a single NVIDIA GeForce RTX 3090 with 24GB memory. The operating system of the machine is Ubuntu 18.04.6. As for software versions, we use Python 3.8, Pytorch 1.11.0, and CUDA 11.4. To alleviate the influence of randomness, we repeat each method 10 times and report the statistical characteristics. The hyper-parameters of baselines are set according to the original paper if available. We use Optuna Akiba et al. (2019) to implement hyperparameters search. Following the above principles, we present the results of two data partitioning as follows.

Table 11: The results of test accuracy based on CiteSeer and Squirrel by implementing the strcuture Non-IID: mean accuracy $\pm$ standard deviation. The best results are shown in bold.

| | CiteSeer | | | Squirrel | | |
| Non-IID | Client3 | Client5 | Client10 | Client3 | Client5 | Client10 |
|---|---|---|---|---|---|---|
| FedMLP | 69.58±0.65 | 71.31±1.06 | 70.92±1.36 | 30.59±0.44 | 30.42±0.46 | 30.27±0.97 |
| FedSGC | 83.12±0.25 | 46.84±0.05 | 49.52±0.19 | 39.06±0.64 | 28.44±0.67 | 36.95±0.23 |
| FedNLGNN | 81.35±0.97 | 70.78±0.98 | 70.17±0.93 | 46.82±2.58 | 37.85±2.17 | 42.33±1.77 |
| FedGGCN | 80.72±0.52 | 70.90±1.54 | 70.83±1.14 | 53.35±0.95 | 43.59±0.90 | 38.22±0.82 |
| FedGL | 82.91±0.99 | 69.28±0.70 | 68.12±0.54 | 35.06±1.93 | 32.88±1.51 | 33.85±2.49 |
| GCFL+ | 83.42±0.08 | 51.70±0.43 | 53.73±0.82 | 46.16±0.31 | 31.08±0.46 | 39.90±0.39 |
| FedSage+ | 69.64±0.63 | 72.49±0.70 | 70.92±0.53 | 34.79±1.26 | 33.37±0.89 | 34.42±2.87 |
| w/o HomoKD | 82.62±0.51 | 72.03±0.84 | 70.89±1.41 | 55.06±1.08 | 44.99±1.08 | 45.89±1.97 |
| w/o HeteTA | 83.37±0.28 | 71.15±0.33 | 70.09±0.72 | 54.22±0.87 | 43.48±0.73 | 42.83±1.39 |
| AdaFGL | **84.05±0.49** | **72.95±0.92** | **71.44±1.53** | **55.27±1.13** | **45.55±1.01** | **46.15±2.06** |

Table 12: The results of test accuracy based on PubMed by implementing the community split and structure Non-IID: mean accuracy $\pm$ standard deviation. The best results are shown in bold.

| | PubMed (Community) | | | PubMed (Non-IID) | | |
| | Client3 | Client5 | Client10 | Client3 | Client5 | Client10 |
|---|---|---|---|---|---|---|
| FedMLP | 84.10±0.44 | 81.92±0.70 | 78.26±0.84 | 85.65±0.29 | 84.78±0.18 | 85.29±0.52 |
| FedSGC | 76.37±0.22 | 77.34±0.31 | 74.32±0.10 | 87.05±0.03 | 87.84±0.19 | 59.72±0.11 |
| FedNLGNN | 85.61±0.26 | 84.04±0.40 | 85.10±0.46 | 90.17±0.26 | 91.20±0.16 | 85.11±0.20 |
| FedGGCN | 86.11±0.29 | 86.53±0.52 | 85.67±0.35 | 90.24±0.19 | 92.38±0.13 | 86.59±0.63 |
| FedGL | 84.17±0.28 | 82.46±0.15 | 80.50±0.76 | 92.14±0.21 | 86.16±0.19 | 85.16±0.19 |
| GCFL+ | 86.38±0.11 | 86.38±0.11 | 86.19±0.05 | 89.99±0.01 | 91.88±0.08 | 74.91±0.14 |
| FedSage+ | 85.33±0.43 | 85.17±0.49 | 84.32±0.57 | 86.12±0.37 | 84.85±0.17 | 86.93±0.56 |
| w/o HomoKD | 84.83±0.21 | 85.17±0.30 | 84.99±0.32 | 92.47±0.08 | 93.17±0.08 | 87.87±0.42 |
| w/o HeteTA | 86.49±0.26 | 86.69±0.35 | 86.01±0.38 | 94.32±0.04 | 93.98±0.04 | 86.23±0.25 |
| AdaFGL | **86.49±0.26** | **86.69±0.35** | **86.49±0.47** | **94.55±0.09** | **95.45±0.07** | **88.39±0.37** |

