# OpenReview forum: "A New Paradigm for Federated Structure Non-IID Subgraph Learning"
_ICLR.cc/2023/Conference — Submitted to ICLR 2023_

### Official Review · Reviewer_45Ek · 2022-10-20

**Confidence:** 3
**Correctness:** 3
**Technical Novelty And Significance:** 2
**Empirical Novelty And Significance:** 2
**Recommendation:** 5

**Clarity, Quality, Novelty And Reproducibility:**

As what I pointed out in weakness 1, there are some confusing points, which makes it hard to follow.

The presentation is poor. The discussion of related work seems to be comprehensive enough, imo. The experiments are designed to answer these five research questions, which are related to the core scientific question of this paper. Thus, I think the quality of the empirical studies in this paper is satisfactory. I guess the authors just rushed to prepare this submission, as the references are not unified in their styles. For example, some KDD22 papers have the venue, but FS-G (also a KDD22 paper) does not.

The HCS is novel to me, which must be helpful for estimating homophily level under the semi-supervised setting. The idea of judging whether message propagation is helpful or harmful has been extensively studied, where high-pass and low-pass (may be also identity) filters are often combined in various ways to handle both homophily and heterophily. The personalization scheme in this paper seems to be novel, but I have not fully understand its details due to the poor clarity.

It seems that the experiments can be easily reproduced, as all the datasets and baselines are open-sourced.

**Strength And Weaknesses:**

Strengths:
1. The proposed paradigm is well motivated, where the structure non-iidness has been discussed in FS-G but has not been well addressed before.
2. As the node classification task is often a semi-supervised learning setting, the proposed homogeneity confidence score (HCS) is interesting and tends to be helpful. It is novel to me.

Weaknesses:
1. It is difficult for me to follow the story due to its poor presentation. Specifically, I cannot find the definition of some important matrix such as $X_{\text{global}}$. Moreover, the bound Eq. 6 and the theorem are very confusing. What is the relationship between base predictor and the HCS, especially considering that MLP does not depend on graph structure? What are the trainable parameters at all? Among them, which are client-wise? In Sec. 3.2, "base predictor to embed local subgraph nodes into the global knowledge space" is confusing. The author just analyzed its error bound in Eq. 6.

**Summary Of The Paper:**

This paper proposes a novel federated learning (FL) paradigm, AdaFGL, for subgraph learning (i.e., each client holds a subgraph and considers node classification or link prediction task). AdaFGL is designed to handle the structure non-iidness, a graph unique non-iidness issue in FL. The authors conducted extensive empirical studies, which show that AdaFGL outperforms SOTA federated graph learning algorithms under both community-based and non-iid structure splits.

**Summary Of The Review:**

I admit that the motivation in this paper is natural and the proposed paradigm seems to be novel. However, due to the weakness in terms of clarity, I cannot fully understand its idea. Although this paper's empirical studies are comprehensive, I still think it has not been above the bar of ICLR. If the authors are willing to giving further explanation, I am likely to change my opinion.

---

### Official Review · Reviewer_fd7h · 2022-10-24

**Confidence:** 4
**Correctness:** 2
**Technical Novelty And Significance:** 3
**Empirical Novelty And Significance:** 3
**Recommendation:** 3

**Clarity, Quality, Novelty And Reproducibility:**

### Clarity
* The results in Figure 2 left are not explained well. What do the colors and numbers denote in Figure 2? How to calculate them?
* Where is the non-params label propagation in Equation (5) used for? Why is it necessary? Based on my understanding, the label propagation is defined, however, this is not used in the proposed AdaFGL.
* I appreciate that the authors make effort to analyze the error bound of the proposed AdaFGL theoretically. However, it is unclear why the approximated error bound for the proposed AdaFGL is necessary. It seems they do not lower the error bound of the existing FL methods, thus are they necessary?
* $X_{global}$, used in Equation (8), is not defined anywhere else. Where did this term come from? Also, it is unclear how to obtain the global embedding, and why it is not changed during FL.
* The results in Figure 5 are not clearly described. The authors explain that there are 10 clients, however, the x- and y-axises indicate the nodes, and there are no explanations which nodes are belong to which clients. Also, what is the meaning of darker blue color in Figure 5? How to calculate such the color?

### Quality
* Few major claims should be tone-downed. In particular, in abstract, the authors claim that covariance-shift challenges occur in the structure Non-IID setting. However, in the existing community-based split scenarios for subgraph FL, since different communities have different properties (i.e., different clients have different properties), the same covariance-shift challenges occur in this community-based split setting as well.

### Novelty
* The consideration of heterophily for subgraph FL is novel.

### Reproducibility
* The reproducibility is high, since the authors provide the source code.

**Strength And Weaknesses:**

### Strengths
* The idea of considering both the homophily and heterophily of graph-structured data for subgraph FL is novel and interesting.
* The proposed AdaFGL outperforms other graph FL baselines.

### Weaknesses
* The experimental setups for heterogeneous assumption are problematic. In particular, the authors use the Dirichlet process (He et al., 2020) for graph-structured data, to simulate the heterogeneous scenarios. However, this Dirichlet process is not suitable for making heterogeneous subgraph FL. This is because, when the number of clients is relatively large (e.g., 10), the number of edges is often smaller than the number of nodes, which is not realistic. Also, when the number of clients is relatively small (e.g., 3), there are no obvious heterogeneous patterns in the partitioned graphs: the data homogeneity of heterogeneous scenarios is similar to the homogeneous scenarios. Those results are reported in Table 13-22, and, based on that, I don't think the evaluation for heterogeneous assumption is correctly done.
* The usage of global data is not convincing, and it is not applicable to realistic FL. In FL, data sharing is not possible, therefore, we cannot obtain the global graph consisting of all nodes distributed to all clients. However, the authors propose the FL framework that leverages the global graph. How can we obtain the global graph, when each subgraph cannot share its subgraph to others under the FL assumption? Also, comparing the proposed AdaFGL that uses global graph against other subgraph FL baselines that do not use global graph looks unfair.
* There are many unclear major-claims, which should be clarified. See Clarity in the below.

**Summary Of The Paper:**

This paper investigates the Non-IID problems of subgraph Federated Learning (FL), where the authors consider both the well-known homogeneity assumption and the under-explored heterogeneity assumption. To tackle those two problems, the authors propose the global knowledge extractor that uses global data to extract features on the global graph, and the adaptive propagation modules that combine global embeddings and locally updated features for local node representations. The authors verify the proposed method, called AdaFGL, on both homogeneous and heterogeneous graph datasets, showing the effectiveness of their AdaFGL.

**Summary Of The Review:**

The idea of considering both homophily and heterophily of graph-structured data for subgraph FL is novel, however, there are many unclear claims and the evaluation setup is not convincing. Thus, I cannot recommend the acceptance.

---

### Official Review · Reviewer_3ztS · 2022-10-31

**Confidence:** 3
**Correctness:** 3
**Technical Novelty And Significance:** 2
**Empirical Novelty And Significance:** 3
**Recommendation:** 6

**Clarity, Quality, Novelty And Reproducibility:**

The writing is quite clear. The setup is new while the proposed method is trivial.

**Strength And Weaknesses:**

Weaknesses:
1. How can you justify the correctness of your structure Non-IID assumption? In the current paper, the authors only provided experimental results on the ideal split dataset according to their assumption.
2. In Table 3, the authors claim that the community split only provides a homogeneous distribution over different clients. The reported results show that AdaFGL obtains *totally the same performance* as the method w/o HeteTA on the Cora dataset. Interestingly, AdaFGL gets *the same performance* as the method w/o HomoKD on the Chameleon dataset. Please check your experimental results or put more effort into providing a reasonable explanation.

**Summary Of The Paper:**

Instead of taking the community split method used in the previous federated graph learning trend, this paper introduces a new heterogeneous graph split method named structure non-iid and designs a new framework called AdaFGL to deal with the new problem. In the method, a HomoKD part is used for propagating the homogeneity message and a HeteTA part is leveraged to fit with the graph heterogeneity. Some experiments are provided to verify the effectiveness of the proposed method.

**Summary Of The Review:**

Learning GNN from heterogeneous data in a federated setup is rather a novel and interesting topic. The authors provide a novel Non-IID setting that is new and different from the previous work. I am not sure about the soundness of this assumption, and the authors only provide datasets split according to this assumption.

---

### Decision · Program_Chairs · 2023-01-20

**Decision:**

Reject

**Justification For Why Not Higher Score:**

experimental evaluations have problems and the presentation is not good.

**Justification For Why Not Lower Score:**

n/a

**Metareview: Summary, Strengths And Weaknesses:**

This paper studies the federated graph learning problem under the non-iid setting. In particular, this paper assumes structure non-iid of the subgraphs and proposes an adaptive federated graph learning method to incorporate adaptive propagation mechanisms into the learning process.

While the general idea of heterogeneity is interesting, there are some crucial concerns on the technical quality and presentation. First, the experimental setup on heterogeneity is problematic. The number of edges is larger than the number of edges, which makes the graph too sparse and not realistic. The statistics of the data is not clearly described. Also, the presentation of this paper is not good, which makes it hard to understand the technical details. Given these concerns, I would recommend rejection of this paper. Although we think the paper is not ready for ICLR in this round, we believe that this paper would be a strong one if the experimental setup issues could be addressed and the presentation could be improved.